# Tagged-Informed Prior for Motion Quantification in Cine CMR Using Implicit Neural Representations

**Laura Alvarez-Florez**[1,2,3,6]                    L.ALVAREZFLOREZ@AMSTERDAMUMC.NL

**Soufiane Ben Haddou**[1,2,3,6]                    S.BENHADDOU@AMSTERDAMUMC.NL

**Fleur V. Y. Tjong**[4,6]                    F.V.TJONG@AMSTERDAMUMC.NL

**Ivana Išgum**[1,5,6,7]                    ISGUM.IVANA@MAYO.EDU

[1] *Department of Biomedical Engineering and Physics, Amsterdam University Medical Center, The Netherlands*

[2] *Informatics Institute, University of Amsterdam, The Netherlands*

[3] *Quantitative Healthcare Analysis group, University of Amsterdam, The Netherlands*

[4] *Heart Center, Department of Clinical and Experimental Cardiology, Amsterdam University Medical Center, The Netherlands*

[5] *Department of Radiology and Nuclear Medicine, Amsterdam University Medical Center, The Netherlands*

[6] *Amsterdam Cardiovascular Sciences, Amsterdam, The Netherlands*

[7] *Department of Radiology, Mayo Clinic, Rochester, United States of America*

**Editors:** Accepted for publication at MIDL 2026

## Abstract

Accurate quantification of myocardial motion from cine cardiac magnetic resonance (CMR) is essential for assessing cardiac function. Although tagged CMR provides high-fidelity measurements of myocardial deformation, its longer acquisition time limits routine clinical use, making cine CMR motion estimation the more widely applicable approach. Implicit neural representations (INRs) offer a promising framework for cine-based motion estimation by modelling cardiac motion as a continuous spatio-temporal function. However, they require subject-specific optimisation and are sensitive to initialization, leading to slow convergence. Furthermore, optimisation from random initialization can lead to large number of solutions that may not guarantee biomechanically plausible motion. To address these limitations, we propose a strategy to improve and accelerate INR-based registration of cine CMR by leveraging a population-level prior derived from tagged CMR data. First, we train subject-specific INRs on the tagged cine dataset to encode characteristic myocardial deformation patterns. Second, we aggregate their parameters across subjects to form a tagged-informed population prior. Third, we use this prior initialization to warm-start the optimization of cine INRs. The resulting prior provides a physiologically meaningful starting point for cine-only INR optimisation, reducing the search space and promoting more realistic cardiac motion. We develop and test the method on the UK Biobank. Compared with standard initialization, the proposed prior enables the INR to reach near-optimal performance using only half as many optimisation steps, achieving a 4% improvement in Dice and a 15% reduction in Hausdorff distance. These gains also translate to a test set of 855 subjects from a different institution, encompassing different pathologies, where the prior yields smoother and more physiologically plausible strain curves. The code for this research is publicly available. [1]

---

1. https://github.com/qurAI-amsterdam/tagged-prior-inr

**Keywords:** Cardiac MRI, Tagged CMR, Implicit Neural Representations, Image Registration, Cardiac Motion Quantification, Prior Initialization

## 1. Introduction

Characterizing myocardial motion is a key component in the assessment of cardiovascular disease. Quantitative descriptors such as ejection fraction and myocardial strain provide insight into global and regional contractility, supporting risk stratification and treatment planning across a range of cardiovascular diseases (Lange and Schuster, 2021; Fudim et al., 2021). Obtaining motion measurements, however, requires precise estimation of myocardial deformation throughout the cardiac cycle, which remains a challenge.

Dedicated sequences such as tagged Cardiac Magnetic Resonance (CMR) provide high-fidelity measurements of regional myocardial deformation and are widely regarded as the reference standard for non-invasive assessment of cardiac motion. However, they require additional scanning time and specialised acquisition protocols, which restrict their use in routine clinical workflows (Ibrahim, 2011). As a result, there is strong interest in deriving motion and strain directly from standard cine CMR routinely acquired in clinic.

Image registration methods have long been used to assess cardiac motion by aligning myocardial anatomy across the cardiac cycle (Zhang et al., 2024). Deep learning (DL) methods build on this principle by learning the registration mapping directly from data, enabling faster inference and robustness to image noise. Most DL-based motion estimation frameworks adopt a registration formulation in which consecutive cine frames are aligned by maximising image similarity (De Vos et al., 2019). Registration methods based on convolutional networks have shown strong performance in this setting (Upendra et al., 2020; Morales et al., 2021; Qin et al., 2020), but they operate on discrete voxel grids, limiting the spatial and temporal continuity of the estimated motion fields. Implicit neural representations (INRs) have recently emerged as an alternative (Wolterink et al., 2022) and have outperformed other registration methods for cardiac motion quantification (López et al., 2023). INRs model motion as a continuous function of spatial and temporal coordinates, fitting a small multilayer perceptron to each subject's cine sequence. Although this yields highly smooth and differentiable displacement fields, a major practical limitation is that INRs must be optimised from scratch for every new subject, resulting in long inference times that hinder clinical usability. Moreover, image registration remains an ill-posed problem, and optimisation guided mainly by image similarity does not guarantee that the estimated cardiac motion is biomechanically plausible.

Recent work suggests that initialization plays a crucial role in addressing these challenges, both by reducing the per-subject optimisation burden and by guiding the network toward more meaningful solutions (Koneputugodage et al., 2025). Initializing the INR closer to a physiologically plausible deformation, instead of starting from random weights, can accelerate convergence and help avoid unrealistic motion estimates. Our prior work demonstrated that, when estimating motion between consecutive cine frames, warm-starting each INR from the previous time point in the cardiac cycle speeds up convergence and improves registration quality (Alvarez-Florez et al., 2024). In this work, we extend this idea by constraining the INR optimisation space using high-fidelity measurements of myocardial deformation encoded in tagged images, with the aim of promoting more biomechanically

plausible motion estimates and faster convergence for cine CMR registration. Due to the lower resolution of the tagged dataset, we first create a dataset that combines these with cine images to increase the through plane resolution. We use a SIREN as our baseline INR architecture, which has demonstrated strong performance in medical image registration tasks (Wolterink et al., 2022). Then, we train these INRs on the tagged and cine data to embed characteristic deformation patterns within their parameters, and aggregate these parameters across subjects to construct a tagged-informed population prior. This prior serves as a physiologically grounded initialization that reduces the optimisation search space, accelerates convergence, and improves the quality of motion estimated from cine CMR. Hence, the main contributions of this work are:

- We introduce the first population-level initialization for INR-based cardiac motion estimation, constructed from tagged and cine CMR. The resulting initialization is lightweight and model-agnostic, with potential to be used in any INR framework for cine motion quantification.

- We show that by using the proposed prior we reduce the INR's required optimization time by half compared to INRs with a conventional initialization scheme.

- We show that the method generalizes in an independent set from a different center that includes patients with different pathologies unseen by the prior.

## 2. Method

We propose a strategy to improve and accelerate INR-based registration of cine CMR by leveraging a population-level prior derived from tagged CMR data (Fig. 1). First, we present the INR method for image registration. Second, we create a tagged-cine dataset and use it to form a population-average prior initialization. Third, we use this prior initialization to warm-start the optimization of cine INRs.

### 2.1. Implicit Neural Representation for Cardiac Motion Quantification

To estimate continuous cardiac motion over the cardiac cycle, we register each cine CMR frame in the cardiac cycle to the final frame of the cine sequence at end-diastole. The registration model is an INR consisting of a multilayer perceptron with sinusoidal activations. (Sitzmann et al., 2020) The multilayer perceptron architecture consist of four input nodes, three fully connected layers of width 256, and three output nodes.

The INR models a continuous four-dimensional displacement field over space and time. Given a spatial location $c = (x, y, z)$ and a temporal coordinate $t$, the network predicts the displacement vector $u$:

$$u(c, t) = f_\theta(c, t), \tag{1}$$

where $f$ denotes the INR with trainable parameters $\theta$. This formulation provides a smooth and differentiable representation of cardiac motion that enables estimation of displacement at arbitrary spatial locations and time points (López et al., 2023).

Prior to training, each cine volume is cropped to a segmentation-derived bounding box around the heart, and resampled into a canonical orientation to ensure spatial consistency

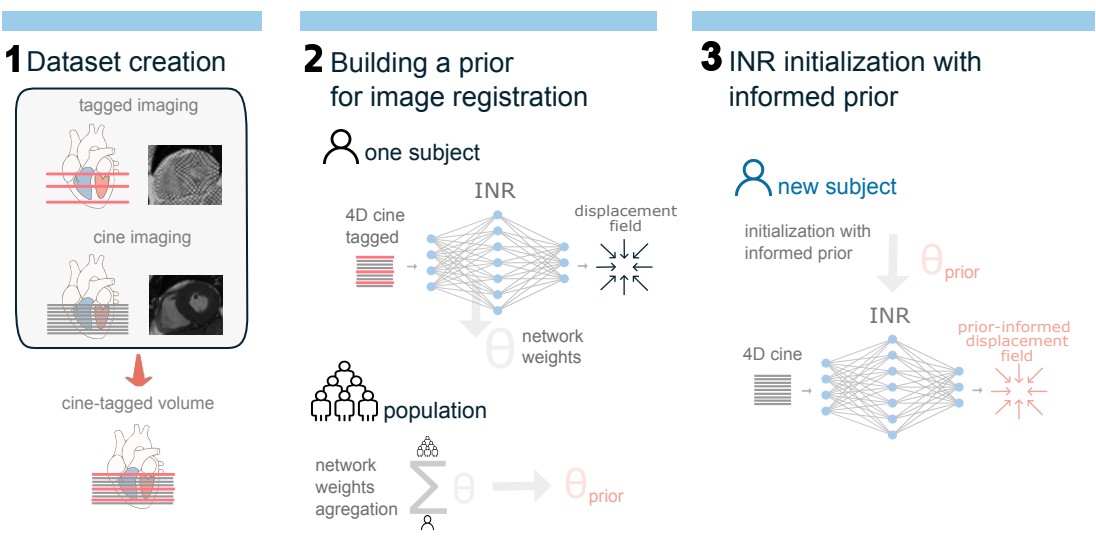

Figure 1: Graphical representation of the proposed tagged-informed registration prior.

across subjects. The input coordinates are normalized to $[-1, 1]$ and uniformly sampled from the cine volume in batches of 10,000 points. The resulting deformation field $\phi$ represented by the network is expressed through the transformation:

$$\phi(c, t) = c + u(c, t), \tag{2}$$

which maps coordinates from time $t$ into the reference frame.

During training, the moving cine frames $I^t$ are warped using the predicted transformation to obtain the corresponding warped frame $\hat{I}^t$ at time $t$:

$$\hat{I}^t(c) = I^t\left(\phi(c, t)\right). \tag{3}$$

The warped image $\hat{I}^t$ is then compared with the reference $I^{\text{ref}}$ using a normalized cross-correlation (NCC) image similarity loss. To encourage physiologically plausible motion and volume preservation, the loss includes a Jacobian-based regularization term within the myocardium:

$$\mathcal{L}_{\text{Jac}} = \frac{1}{N} \sum_c \left[ \left( \det J(c) - 1 \right) + \left( \det J(c)^{-1} - 1 \right) \right], \tag{4}$$

where $J = \nabla \phi$ and $N$ represents the number of sampled coordinates. The total loss is therefore defined as:

$$\mathcal{L}_{\text{registration}} = \mathcal{L}_{\text{NCC}} + \alpha * \mathcal{L}_{\text{Jac}}, \tag{5}$$

where $\alpha$ controls the strength of the deformation regularization.

After optimization, myocardial strain is derived from the displacement field by computing the Lagrangian strain tensor from the spatial gradients of the deformation field $\phi$.

Radial and circumferential strain components are then obtained by projecting the strain tensor onto voxel-wise radial directions, defined from the endocardial surface normals, and circumferential directions, defined as the in-plane orthogonal vectors.

All INRs were trained using the Adam (Kingma and Ba, 2014) optimizer with a learning rate of $1 \times 10^{-5}$. For the Jacobian regularization term, we used a weighting $\alpha$ term of 0.25 for the myocardium and 0.0001 for the rest of the image. The INR architecture employed sinusoidal activation functions with the frequency scaling parameter $\omega_0$ set to 45.

## 2.2. Learning a Population Prior from Tagged CMR

To learn a population prior, we first construct a tagged-cine dataset. Tagged CMR provides explicit information about myocardial deformation, making it well suited for learning a biomechanically informed prior. Unlike cine CMR, which typically consists of a full short-axis stack (12 slices, 3D+time), tagged CMR is limited to three 2D+time slices acquired at basal, mid, and apical levels. To capture the deformation information in tagged images without losing through-plane spatial context, we replace each cine slice into the corresponding spatial location of the paired subject's tagged imaging. The resulting tagged–cine volumes are spatially and temporally aligned, preserving tag-derived deformation while providing the full anatomical context required for training the four-dimensional INRs.

For each subject in the tagged-cine dataset $s = 1, \ldots, N$, we independently optimize an INR with identical architecture and training scheme as described in Section 2.1.

To ensure that differences in the learned parameters primarily reflect subject-specific deformation patterns rather than initialization or training variability, all networks are initialized with the same set of parameters and trained in a shared canonical coordinate system with spatially aligned images and normalized coordinates.

After training, the parameter sets $\theta^{(1)}, \ldots, \theta^{(N)}$ are aggregated to form the population prior. For each parameter tensor, we compute the elementwise mean across each subject:

$$\theta_{\mathrm{avg}} = \frac{1}{N} \sum_{s=1}^{N} \theta^{(s)}. \tag{6}$$

To reduce the influence of outlier parameters caused by noise or optimization instability, we clip each parameter tensor to the 1st and 99th empirical quantiles before averaging. The resulting averaged parameters $\theta_{\mathrm{avg}}$ capture the typical displacement behaviour across the tagged population. When applied to cine registration, $\theta_{\mathrm{avg}}$ is used to initialize the INR, after which the network performs image registration as described in Section 2.1. During optimization, all parameters remain fully trainable. Thus, the prior influences only the starting point of the optimization, not the registration formulation itself.

## 2.3. Dataset

For this study, we used two sources of CMR data. The UK Biobank dataset for prior construction and test, and an independent test set from Amsterdam University Medical Center (AUMC) for testing cross-dataset generalization of the prior.

**UK Biobank dataset.** We selected 1190 participants from the UK Biobank imaging dataset (Petersen et al., 2016) who underwent both cine and tagged CMR acquisitions

during the same visit. We split this data into 890 for the development set, and two test sets. Test set 1 with 200 subjects for cine registration evaluation, and Test set 2 with 100 subjects for determining the optimal prior. Short-axis cine images contain up to 12 slices and 50 temporal frames per sequence. Tagged CMR images consist of three independent 2D+time slices located at basal, mid-ventricular, and apical levels. Segmentations of the left-ventricular (LV) myocardium, LV blood pool (LVBP), and right-ventricular (RV) cavity in cine CMR were generated automatically for this work using a deep learning model (Simonyan and Zisserman, 2014). These segmentations were transferred to the corresponding tagged slices using the affine transformation matrix from the image metadata.

**AUMC dataset.** For independent evaluation, we used a dataset of 855 patients referred for implantable cardioverter-defibrillator (ICD) implantation. The dataset consists of patients suffering from different cardiomyopathies, specifically: hypertrophic cardiomyopathy (HCM), dilated cardiomyopathy (DCM), and ischemic cardiomyopathy (ICM). Each subject underwent a standard short-axis cine CMR examination acquired using balanced steady-state free precession (bSSFP) under breath-hold conditions. A full short-axis stack was available for each patient, typically comprising 10–25 slices with 15–25 temporal frames covering one cardiac cycle. Automated segmentations of the LV, LVBP and RV were obtained using an automated deep-learning segmentation pipeline (Sander et al., 2020). Use of this dataset was approved by the institution's medical ethics committee.

## 2.4. Evaluation

To assess registration performance, we use standard segmentation-based overlap and distance metrics, together with deformation-based measures. All metrics are computed for the LV, RV, and LVBP and reported as the average across structures.

**Segmentation metrics.** We report three widely used measures for the evaluation of image registration through registration: the Dice similarity coefficient, quantifying volumetric overlap; the 95th percentile Hausdorff distance (HD95), capturing boundary alignment; and the average surface distance (ASD), reflecting overall surface agreement. Higher Dice and lower HD95/ASD values indicate better registration performance.

**Deformation metrics.** To characterize the smoothness and biomechanical plausibility of the deformation field, we compute the variability of the Jacobian determinant on the myocardium, $|\det(J) - 1|$, which penalizes local expansions or contractions. Lower values indicate more spatially regular, volume-consistent motion. This metric captures the magnitude of local expansions and contractions irrespective of their sign, and is therefore sensitive to spatial variability and heterogeneity of the deformation field. Additionally, local non-invertible deformations are quantified using the folding ratio, defined as the proportion of voxels with negative Jacobian determinant. A lower folding ratio corresponds to more physically plausible deformations.

**Strain curves.** Quantitative geometric metrics describe local deformation properties, but they do not fully capture the temporal coherence or global physiological behaviour of cardiac motion. To complement these metrics, we examine the derived radial and circumferential strain trajectories, offering additional insight into myocardial mechanics.

Table 1: Registration performance for the UK Biobank Test set 1 (mean ± std).

| Model | Dice (↑) | HD95 (↓) | ASD (↓) | Jacobian Determinant (↓) |
|---|---|---|---|---|
| Baseline | 0.733 ± 0.097 | 1.933 ± 1.063 | 0.536 ± 0.252 | **0.151 ± 0.022** |
| Prior | **0.763 ± 0.073** | **1.642 ± 0.760** | **0.461 ± 0.180** | 0.170 ± 0.024 |

## 3. Experiments and Results

First, we present the quantitative registration performance results on the UK Biobank and AUMC test sets. Second, we perform a qualitative analysis of the strain. Third, we present an ablation demonstrating the optimal prior settings used for our evaluation.

### 3.1. Results of Registration Performance

The cine registration performance was assessed on Test set 1 from the UK Biobank. Quantitative results are summarized in Table 1. The proposed method was compared with a baseline. The baseline consisted of the INR method initialized with Xavier initialization (Glorot and Bengio, 2010). Across all geometric metrics, the prior-initialized model consistently outperformed the baseline. Dice similarity increased by approximately 4%, indicating improved volumetric alignment. Boundary-based metrics also showed substantial gains: HD95 decreased by roughly 15% and ASD by about 14%, reflecting more accurate contour alignment and smoother spatial correspondence.

The folding ratio remained at zero for both models, confirming the absence of topological violations across all test cases. The Jacobian determinant increased (around 12%) when using the prior, reflecting the influence of the strong initialization. To further analyze the behavior of the Jacobian determinant, Figure 6 in Appendix B reports the distribution of subject-level mean Jacobian variability across different cardiac regions. The boxplots show that the increased Jacobian variability observed for the prior-initialized model is not uniform across the image, but predominantly arises in regions that are less constrained during prior construction, such as the RV and non-cardiac background. In contrast, the LV myocardium, where Jacobian strongly regularized, exhibits comparable variability between the prior-initialized and baseline models. These results indicate that the observed increase in Jacobian determinant values reflects region-specific effects of the initialization strategy, with stable behavior in the primary region of interest, the LV myocardium.

In addition, to assess generalization across centers, scanners, and pathologies, we evaluated the model on the AUMC test set. Quantitative results for each cardiomyopathy are reported in Table 2. Across all cardiomyopathies, the prior-initialized INR outperformed the baseline on the segmentation metrics, mirroring the trends observed in the UK Biobank Test set 1. Improvements were most pronounced for the HCM group, where Dice increased by roughly 3% and both HD95 and ASD decreased noticeably. On the other hand, the Jacobian Determinant presented lower values for the baseline method. For the DCM and ICM groups, improvements were more modest but remained consistent across metrics.

Table 2: Registration performance for the AUMC test set (mean ± std).

| Cohort | Model | Dice (↑) | HD95 (↓) | ASD (↓) | Jacobian Determinant (↓) |
|--------|-------|----------|----------|---------|--------------------------|
| HCM | Baseline | $0.752 \pm 0.116$ | $2.329 \pm 2.759$ | $0.619 \pm 1.226$ | $\mathbf{0.124 \pm 0.027}$ |
|     | Prior | $\mathbf{0.777 \pm 0.099}$ | $\mathbf{2.034 \pm 2.639}$ | $\mathbf{0.548 \pm 1.208}$ | $0.135 \pm 0.033$ |
| DCM | Baseline | $0.781 \pm 0.091$ | $2.208 \pm 4.708$ | $0.617 \pm 2.633$ | $\mathbf{0.113 \pm 0.024}$ |
|     | Prior | $\mathbf{0.798 \pm 0.080}$ | $\mathbf{2.042 \pm 4.667}$ | $\mathbf{0.573 \pm 2.592}$ | $0.120 \pm 0.028$ |
| ICM | Baseline | $0.762 \pm 0.119$ | $2.011 \pm 1.890$ | $0.518 \pm 0.517$ | $\mathbf{0.121 \pm 0.027}$ |
|     | Prior | $\mathbf{0.772 \pm 0.133}$ | $\mathbf{1.850 \pm 1.932}$ | $\mathbf{0.494 \pm 0.665}$ | $0.128 \pm 0.030$ |

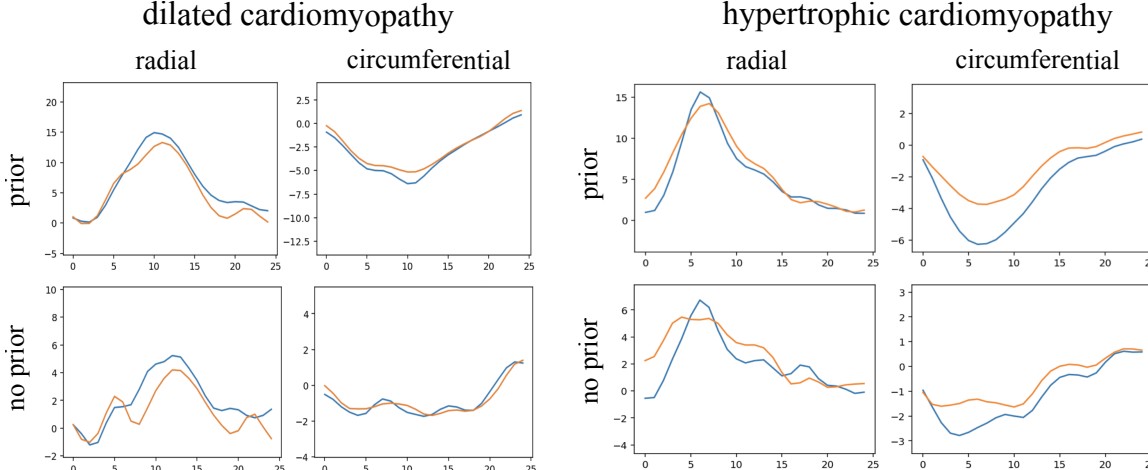

Figure 2: Radial and circumferential strain curves for a patient with dilated cardiomyopathy (DCM, left) and hypertrophic cardiomyopathy (HCM, right). Top row: registration using the tagged-informed INR prior. Bottom row: baseline INR without prior. The prior produces smoother and more physiologically plausible endocardial and epicardial strain trajectories. Endocardium is depicted in color blue and epicardium in orange.

## 3.2. Qualitative strain analysis

We conducted a qualitative evaluation to study the biomechanical plausibility, beyond deformation based metrics, of the derived strain trajectories across models. Figure 2 illustrates the qualitative strain patterns for representative DCM and HCM cases. The prior-initialized INR produces visibly smoother and more physiologically consistent radial and circumferential strain trajectories for both DCM and HCM cases. In particular, the prior reduces noise in the trajectory and preserves the expected systolic peak and diastolic relaxation pattern. Additionally, differences in trajectories between endocardial and epicardial strains are more consistent and plausible.

Table 3: Results on the UK Biobank Test set 2 showing the registration performance (mean ± std) across baseline method and priors derived from different population sizes. N refers to the number of subjects used to create the prior initialization. Arrows indicate whether higher (↑) or lower (↓) is better.

| Model | Dice (↑) | HD95 (↓) | ASD (↓) | Jacobian Determinant (↓) |
|---|---|---|---|---|
| Baseline | $0.732 \pm 0.097$ | $1.969 \pm 1.154$ | $0.544 \pm 0.259$ | $\mathbf{0.148 \pm 0.021}$ |
| Prior N=50 | $\mathbf{0.763 \pm 0.074}$ | $1.678 \pm 0.882$ | $0.466 \pm 0.190$ | $0.169 \pm 0.023$ |
| Prior N=200 | $0.762 \pm 0.074$ | $1.684 \pm 0.882$ | $0.467 \pm 0.189$ | $0.169 \pm 0.023$ |
| Prior N=450 | $\mathbf{0.763 \pm 0.074}$ | $\mathbf{1.677 \pm 0.875}$ | $\mathbf{0.465 \pm 0.189}$ | $0.169 \pm 0.023$ |
| Prior N=890 | $0.762 \pm 0.074$ | $1.686 \pm 0.891$ | $0.466 \pm 0.191$ | $0.169 \pm 0.024$ |

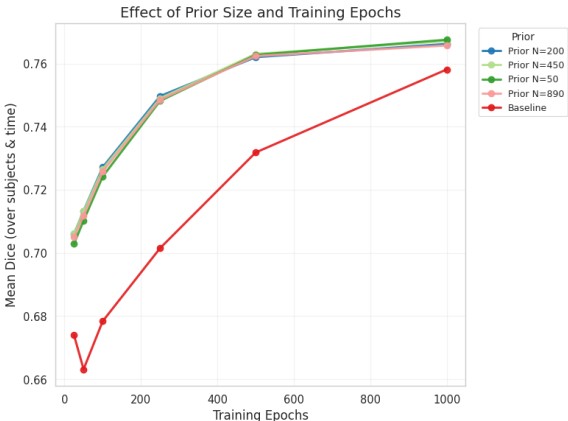

Figure 3: Effect of prior size and training epochs on registration performance, measured by mean Dice on UK Biobank Test set 2. Priors accelerate early performance gains, with only marginal improvements beyond approximately 500 epochs.

### 3.3. Influence of Initialization Strategy

The ablation study evaluates how the size of the population used to construct the initialization prior, and the number of training epochs, affect registration performance. We compared four priors derived from populations of size $N = 50, 200, 450, 890$ against the baseline model presented in the previous sections, and trained each configuration for 20, 50, 250, 500, and 1,000 epochs. This analysis was performed on Test set 2 from the UK biobank.

Figure 3 shows the mean Dice across subjects as a function of training epochs and prior size. All population priors lead to substantially faster performance gains than the baseline, with convergence curves that are nearly identical across prior sizes. This indicates that even small cohorts (e.g., $N = 50$) provide an initialization strong enough to guide the INR effectively, and increasing the population size does not meaningfully influence performance.

### zero-shot comparison

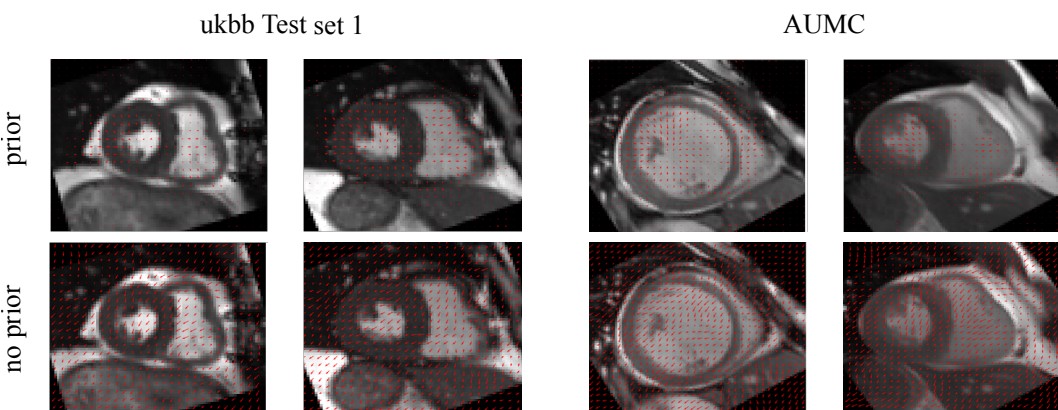

Figure 4: Results showing the differences in the displacement vector fields, before any optimization, with (top) and without (bottom) the proposed prior in a mid ventricle slice at contraction. The two columns on the left show two different patients from the UK Biobank. The columns on the right show a patient with dilated cardiomyopathy and ischaemic cardiomyopathy from our institution, respectively.

The training time analysis shows that after roughly 500 epochs, prior-initialized models reach performance comparable to or exceeding the baseline trained for 1,000 epochs.

To further investigate the effect of the size of the prior, we report registration accuracy across multiple metrics. Table 3 shows the results for different INR models trained for 500 epochs. All prior initializations provide a clear improvement over the baseline, with increases of approximately 3–4 percentage points in Dice, and consistent reductions in HD95 and ASD. The Jacobian determinant metric exhibits slightly higher variability for prior-based models, likely reflecting the stronger initial constraints imposed by the population initialization. The folding ratio, not presented on the table, remains near zero for all methods, indicating that none of the priors introduces folding artefacts. Differences between prior sizes remain insignificant and consistent across metrics.

Finally, to better understand the inductive bias introduced by the proposed population prior, Figure 4 shows the effect of the proposed initialization prior before any optimization is performed. The figure presents displacement vector fields produced by the prior INR and baseline INR before training for subjects from the UK Biobank and the AUMC test set. While these initial fields are not intended to represent accurate motion estimates, the prior-initialized model produces displacement fields that are localized to the myocardium and exhibit more consistent directional patterns broadly reflecting typical cardiac contraction, whereas the baseline initialization results in largely unstructured fields. This qualitative result illustrates that the population prior encodes shared task-related structure at initialization, which is subsequently refined during optimization. To further contextualize this observation, Fig. 5 in Appendix A additionally examines a prior constructed from INRs initialized with different seeds, demonstrating that without alignment in weight space the

resulting zero-shot displacement field collapses to near-zero magnitude, highlighting the importance of an aligned initialization scheme for constructing a meaningful prior.

## 4. Discussion

In this work, we proposed a tagged-informed population prior for INR-based cine CMR registration and demonstrated that it substantially accelerates convergence while improving registration accuracy. By averaging the parameters of subject-specific INRs trained on tagged CMR, we obtained a population template that captures typical myocardial motion patterns and provides a strong initialization for subsequent cine registration. To our knowledge, this is the first study to exploit tagged CMR in this way, bridging its informative myocardial deformation to more widely available cine imaging.

The evaluation showed that the proposed prior consistently outperformed the baseline INR initialized with standard weights on the segmentation metrics. Convergence curves revealed that prior-based models reached near-optimal performance after about 500 epochs, whereas the baseline required roughly twice as many epochs to reach comparable accuracy. These results suggest that the prior effectively guides optimization into a favourable region of the parameter space, reducing the burden of per-subject INR training. Future work should explore its practical implications, including whether such reductions in computation time can help bridge the gap toward real-time cardiac motion estimation in clinical workflows.

The results generalized well to the test dataset, composed of cine CMR images from ICD patients, a different population than the one used to construct the prior, acquired at a different institution. Improvements were particularly consistent for HCM, suggesting that the tagged-informed prior can transfer beyond the population from which it was learned. In contrast, gains were more modest for DCM and ICM, whose deformation patterns are substantially altered by global or regional hypokinesia and scar tissue. Since the prior was derived from a general population, its inductive bias is naturally better aligned with preserved or hyperdynamic contraction, such as in HCM, and less representative of the severely abnormal motion commonly seen in DCM and ICM. These findings indicate that the behavior of the prior is sensitive to the composition of the tagged cohort used during its construction. While a general population prior offers robustness and broad applicability, alternative strategies such as pathology-stratified priors or adaptive weighting schemes that emphasize specific motion patterns may better capture disease-specific mechanics without sacrificing generalization.

While the quantitative metrics evaluated in this study capture important aspects of registration performance, our qualitative analysis of myocardial strain revealed additional benefits of the proposed prior. The prior-initialized INRs produced smoother, more physiologically plausible strain trajectories, with clearer systolic peaks and more consistent separation between endocardial and epicardial layers. These findings suggest that the advantages of the proposed initialization extend beyond geometric accuracy, and influence the biomechanical fidelity of the estimated motion fields. Future work should investigate these effects in greater depth, including a characterization of strain-derived biomarkers and their potential clinical utility in detecting subtle abnormalities in myocardial mechanics. Additionally, a direct quantitative comparison with tagged-derived strain measurements, where available,

would further strengthen the validation of biomechanical accuracy of the derived strain and is an important direction for future work.

Interestingly, the ablation study revealed that performance improvements saturated quickly with the size of the population used to build the cine tagged-informed prior. Even a prior constructed from only 50 subjects yielded substantial gains over the baseline. This robustness is encouraging from a practical standpoint, as centers with limited images available may be able to construct an effective prior. Future work could explore whether the number of subjects used to build the population prior can be further reduced and how such changes may affect the quality of the registration.

A final consideration relates to how the population prior is constructed by averaging the parameters of independently trained INRs. While neural networks can exhibit permutation symmetries in theory, our results indicate that, under the task constraints and architectural choices considered in this work, the resulting prior provides a stable and informative initialization. This observation is consistent with prior work showing that parameter averaging is effective when models remain aligned in weight space (Wortsman et al., 2022). Nevertheless, alternative approaches for learning shared initializations may further enhance robustness. In particular, meta-learning–based strategies are promising, as they explicitly optimize for rapid adaptation across subjects, but are challenging to apply directly in this setting due to the high dimensionality of cine CMR images and associated computational costs. An interesting direction for future work is therefore to combine a population prior with meta-learning, using it as a strong starting point to guide optimization toward physiologically meaningful motion patterns.

## 5. Conclusion

We introduced a cine tagged population prior for INR-based cine CMR registration. We showed that the proposed prior provides a strong and effective initialization, accelerating training, and improving registration performance and strain estimation. Moreover, we demonstrated that the learned prior generalizes across institutions, offering a practical path towards making INR-based cardiac motion analysis more efficient and scalable.

## Acknowledgments

This research has been conducted using the UK Biobank Resource under application number 24711.

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

## Appendix A. Analysis of different initialization schemes

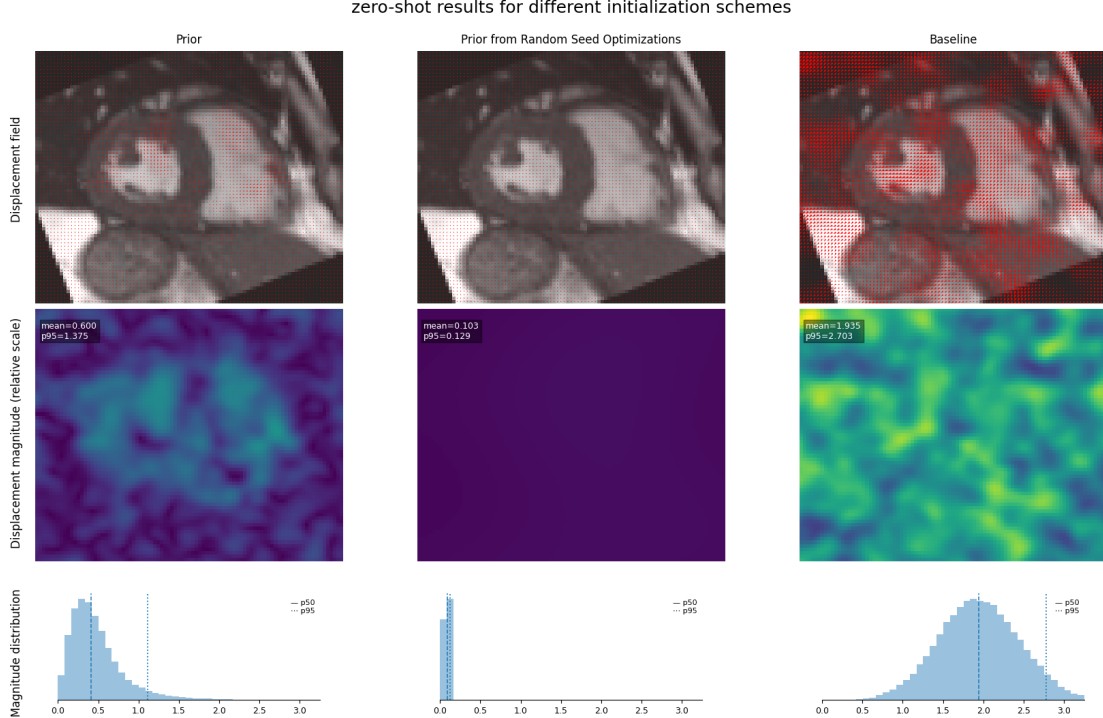

Figure 5: Displacement fields comparison at zero-shot between different initialization schemes. The top row shows the predicted displacement vectors overlaid on the cine image, the middle row shows the corresponding displacement magnitude, and the bottom row shows the distribution of displacement magnitudes for the image volume. All INRs use the same architecture and training setup; both priors are constructed by averaging 450 trained per-subject INRs and differ only in the initialization seed. For the proposed prior (left), all INRs are trained from the same initialization seed, whereas for the different seed prior (middle), each INR used to construct the prior is trained from a different random seed. The different seed prior produces a near-zero displacement field, while the proposed prior retains coherent motion patterns and the baseline initialization (right) exhibits largely unstructured noise.

## Appendix B. Analysis of Jacobian determinant variability

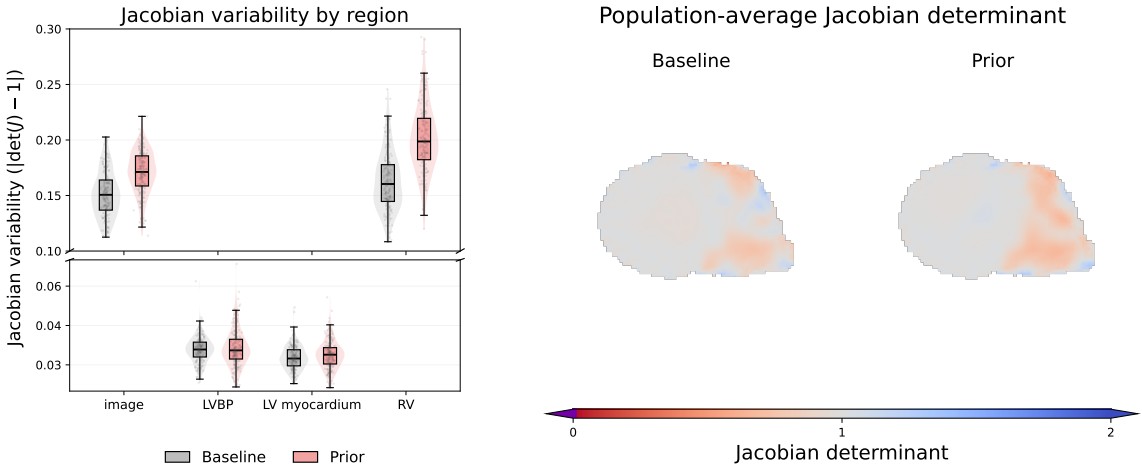

Figure 6: Analysis of Jacobian determinant variability for Test set 1. **(a)** Distribution of subject-level mean Jacobian variability computed over different cardiac structures for baseline and prior-initialized registrations. **(b)** Jacobian determinant maps for Test set 1 computed at a representative mid-ventricular slice and end-systolic cardiac phase. Folding of the registration, defined as negative values, is represented in color purple.

