# OpenReview forum: "Tagged-Informed Prior for Motion Quantification in Cine CMR Using Implicit Neural Representations"
_MIDL.io/2026/Conference — MIDL 2026 Poster_

### Official Review · Reviewer_Xcm3 · 2025-12-28

**Confidence:** 3
**Preliminary Rating:** 4

**Summary:**

This paper proposes a tagged-informed population prior to accelerate and stabilize implicit neural representation (INR)–based cardiac motion estimation from cine CMR. The method constructs a physiologically meaningful initialization by averaging parameters of subject-specific INRs trained on tagged CMR, and uses this prior to warm-start cine-only INR optimization. The authors evaluate the approach on a large UK Biobank cohort and an independent multi-pathology dataset from another institution, demonstrating faster convergence, modest improvements in geometric registration metrics, and smoother, more plausible myocardial strain trajectories.

**Strengths:**

The paper’s main strength lies in its clear and well-motivated use of tagged CMR as a source of biomechanical prior knowledge, which is then transferred to cine-only motion estimation. The idea of aggregating subject-specific INR parameters into a population-level initialization is conceptually simple and well aligned with the cardiac motion estimation.

The method is trained and tested on a large UK Biobank cohort and further validated on an independent external dataset from a different institution with multiple cardiomyopathies.

**Weaknesses:**

While the proposed framework is well motivated and carefully evaluated, several aspects could be strengthened to improve clarity and impact. First, although tagged CMR is used to construct a population prior, the relationship between tagged-derived deformation and cine-derived motion accuracy is not quantitatively assessed. Incorporating a direct comparison between cine-based strain estimates and tagged-based strain, where available, would provide stronger evidence that the prior improves biomechanical fidelity rather than only geometric alignment.

Second, the analysis of deformation regularity shows that the Jacobian determinant variability slightly increases when using the prior. This effect is reported but not fully interpreted. A more detailed discussion or additional experiments examining how this trade-off influences motion plausibility and strain reliability would help clarify the implications of the prior on deformation behavior.

Third, the prior is constructed from a general population, yet the results indicate variable gains across different cardiomyopathies. Exploring pathology-stratified priors or adaptive weighting schemes could help determine whether disease-specific motion patterns can be better captured without sacrificing generalization.

**Detailed Comments:**

The paper would benefit from a clearer discussion of how sensitive the prior is to the choice of tagged cohort (e.g., healthy vs. pathological dominance).

Quantitative evaluation of strain accuracy against tagged-derived strain, where available, would further strengthen the claims.

**Justification Of The Preliminary Rating:**

The paper addresses a practical limitation of INR-based cardiac motion estimation by introducing a tagged-informed population prior that improves optimization stability and convergence. The approach is technically sound and supported by thorough experiments on a large public dataset as well as an independent external cohort, which supports its generalizability. While the contribution focuses on initialization rather than introducing a new motion model, the improvements are consistent and relevant for computationally intensive INR frameworks. Overall, the work is solid, clearly presented, and of practical interest, which supports a weak accept.

**Questions To Address In The Rebuttal:**

Details in the Weaknesses and Detailed Comments

---

> ### Author Response · Authors · 2026-01-25
> **Response to Reviewer Xcm3**
>
> We thank the reviewer for their valuable comments. We have addressed them point by point below.
>
> **W1 & DC2 (quantitative comparison with tagged-derived strain):**
>
> We agree that a direct quantitative comparison between cine-derived strain estimates and tagged-derived strain would provide stronger evidence that the proposed prior improves biomechanical fidelity. Unfortunately, such a comparison is not straightforward, as tagged-derived strain measurements were not available as ground truth in our datasets. In the absence of these, we focused on complementary evidence. We reported qualitative strain analyses showing smoother strain trajectories, clearer systolic peaks, and more consistent endocardial–epicardial separation for prior-initialized models compared to the baseline. These suggest that the proposed prior influences not only geometric alignment but also the biomechanical coherence of the estimated motion. Nevertheless, we fully agree that quantitative validation against tagged-derived strain would be a valuable next step. We have therefore explicitly added this limitation and future direction to the Discussion section, as such an analysis would strengthen the conclusions of this work **(Section 4, paragraph 4)**:
>
> *“Additionally, a direct quantitative comparison with tagged-derived strain measurements, where available, would further strengthen the validation of biomechanical accuracy of the derived strain and is an important direction for future work.”*
>
>
> **W2 (Jacobian determinant):**
>
> We thank the reviewer for highlighting this point and agree that the observed increase in Jacobian determinant variability requires careful interpretation. To better understand this effect, we performed an additional region-specific analysis of the Jacobian determinant and added corresponding visualizations to the revised manuscript.
>
> As shown in **Figure 3**, the increase in Jacobian variability associated with the prior is not uniform across the image. Instead, it predominantly arises in regions that are less explicitly regularized during prior construction, such as the RV and non-cardiac background. In contrast, the LV myocardium, which is constrained during prior construction through a Jacobian regularization term, maintains comparable behaviour, whereas less constraint regions such as the RV and non-cardiac background exhibit increased variability (importantly, the folding ratio of the registration remains 0 for both methods across the image). This analysis is now presented in **Section 3.1, paragraph 2** of the revised manuscript.
>
>
> **W3 & DL1 (pathology specific prior):**
>
> We agree with the reviewer that the observed variability in performance gains across different cardiomyopathies highlight an important limitation of constructing the prior from a general population.
>
> In this work, we intentionally focused on a general population prior to assess whether a single, broadly applicable initialization could improve convergence and performance across diverse cases without sacrificing generalization. The results suggest that this is possible, but as pointed out, it also reveals opportunities for improvement. Pathology-stratified priors or adaptive weighting schemes represent promising extensions that could better capture disease-specific motion patterns while retaining the benefits of population-based initialization. We have added a clearer discussion of the sensitivity of the prior to cohort composition and these potential extensions as future work in the **Discussion section, paragraph 3**:
>
> *“[...] Since the prior was derived from a general population, its inductive bias is naturally better aligned with preserved or hyperdynamic contraction, such as in HCM, and less representative of the severely abnormal motion commonly seen in DCM and ICM. These findings indicate that the behavior of the prior is sensitive to the composition of the tagged cohort used during its construction. While a general population prior offers robustness and broad applicability, alternative strategies such as pathology-stratified priors or adaptive weighting schemes that emphasize specific motion patterns may better capture disease-specific mechanics without sacrificing generalization.”*

---

### Official Review · Reviewer_hqLM · 2026-01-02

**Confidence:** 5
**Preliminary Rating:** 3
**Final Rating:** 4

**Summary:**

This paper proposes an initialization strategy for implicit neural representation (INR) based cine CMR registration. The authors propose to (1) train separate networks per-patient, (2) aggregate a prior-informed initialization by averaging the weights and (3) apply this initialization to unseen cases. The authors demonstrate improved training speed and performance, with their prior effectively generalizing across institutions.

**Strengths:**

The paper is generally easy to read and addresses an interesting problem. Its main strengths are:
- INR-based method recently gained popularity in the medical image computing community and have widely been applied to various registration problems. Finding ways to speed up the training process therefore is relevant and interesting.
- The idea of distilling biomechanically meaningful motion patterns from tagged CMR into a reusable prior is creative and addresses a real limitation of INRs.
- The method is fairly simple, but well presented and easy to follow. Key values, as the population size are determined in ablation studies and the evaluation of the method goes beyond simple registration metrics.
- The paper has a solid experimental design. It includes two datasets UK Biobank & AUMC, with the test set containing pathological cases unseen during prior construction.

**Weaknesses:**

While this is a generally interesting paper, it has several weaknesses:
- My biggest concern is the way the prior is constructed. MLPs have inherent permutation symmetries. For any hidden layer, you can permute the neurons (along with their incoming and outgoing weights) and obtain a functionally identical network. For a layer with $n$ neurons, there are $n!$ equivalent parameter configurations representing the same function.
When the authors train subject-specific INRs independently and then average their parameters, they're averaging across potentially misaligned weight spaces (a problem widely known in the model merging community). Consider two networks that have learned similar cardiac motion patterns but with permuted hidden units. Averaging their weights doesn't produce a meaningful interpolation of the learned functions, it produces essentially noise. While the authors address some of these concerns by using similar initializations and random seeds, this limitation should be discussed in the paper. I think that meta-learning an initialization (as in e.g. this paper: https://arxiv.org/abs/2012.02189) would lead to much better results.

- The paper only compares prior-informed INR to Xavier initialized INR. A comparison to other registration methods (like VoxelMorph or TransMorph) is missing, which makes it hard to judge the actual contribution in comparison to other learning-based methods.
- I was missing an explanation for the consistently worse Jacobian determinant when using the prior. Is there a specific reason or at least an intuition on why this is the case?
- The paper misses important training hyper parameters as the used optimizer or learning rate. There is also no $\alpha$ value reported for the loss function, and no $\omega$ value is given for the SIREN.
- I find the mathematical notation a bit weird, as e.g. $x=(x,y,z)$ reuses $x$. I'd consider rewriting it to e.g. $c=(x,y,z)$ with $u(c,t) = f_{\theta}(c,t)$ later.

**Detailed Comments:**

I have no further comments beyond the ones made in the Weaknesses section.

**Justification Of Final Rating:**

The rebuttal has addressed most of my questions, and my primary concern regarding the construction of the prior has been largely clarified. The arguments and additional experiments presented are more convincing now. I am happy to raise my score to "Weak accept".

**Justification Of The Preliminary Rating:**

This is a good and interesting paper. However, I still have some major concerns that must be addressed before this work can potentially be published. I'm more than happy to raise my score if the authors can clarify these issues, but for now this is a borderline paper for me.

**Questions To Address In The Rebuttal:**

I'd like the authors to focus their answer on my main concern (the way they build their initialization). If possible I'd really like to see a comparison to meta-learned initializations with MAML. The authors should at least give a rational for why their method should still work that goes beyond empirical studies.

---

> ### Author Response · Authors · 2026-01-25
> **Response to Reviewer hqLM**
>
> We thank the reviewer for the critical assessment and for the very clear explanation of the main concerns. We have addressed them point by point below.
>
> **W1.1 (how the prior is constructed):**
>
> We agree that, in general, averaging parameters of independently trained MLPs is problematic due to permutation symmetries in hidden layers, and that averaging weights across misaligned representations can lead to uninterpretable results. This limitation is well known, and we agree that it merits explicit discussion. That said, we argue that in our specific setting the proposed prior retains meaningful shared structure rather than collapsing to noise. This is supported by both the design of the learning task and empirical observations:
>
> **(1) Task structure and coordinate standardization.** All subject-specific INRs used to construct the prior are trained in a shared canonical coordinate system. Prior to training, images are spatially aligned and oriented to a common reference, and spatial coordinates are normalized. While inter-subject anatomical variability remains, the global geometry remains mostly consistent across patients. Consequently, the networks are not learning arbitrary functions, but similar mappings representing cardiac motion. This task alignment reduces the degrees of freedom for independently trained networks.
>
> **(2) Network design and optimization.** We employ sinusoidal activation functions together with a consistent initialization scheme and identical training hyperparameters across subjects. Sinusoidal activations induce smooth, continuous derivatives, which promote stable gradient-based optimization and reduce sensitivity to small perturbations. We hypothesize these constrain both the magnitude and role of individual neurons. Intuitively, this would mean that a small change in function parameters (the weights of the INR) would result in a small change of function outcome.
>
> Moreover, the initialization tightly controls the frequency content and scale of the learned parameters. All INRs used to construct the prior are initialized with the same parameters. Because of this, given the sinusoidal activation properties and task structure, the independently trained INRs tend to converge to similar function parameters. As a result, parameter configuration will be aligned across the INRs, which means that the averaging of the function parameters preserves meaningful structure. Essentially, we argue that the averaging of the parameters preserves the major training data modes, which makes the prior give an informed starting point for the image registration. This behavior is consistent with prior observations in literature, where interpolation and averaging in INR weight space have been shown to preserve meaningful structure for related signals (https://arxiv.org/pdf/2302.05438, see Figure 31 in appendix K).
>
> **(3) Added “zero-shot” empirical validation.** To gain further insight into whether the proposed prior collapses to noise, we evaluated the prior *before any optimization* and compared it to the baseline initialized INR. As shown in **Figure 5** of the revised manuscript, the prior-initialized network produces displacement fields that are spatially localized to the cardiac region and exhibit a consistent directional pattern across subjects, whereas the random initialization yields unstructured outputs.
>
> We have expanded this motivation on **Section 2.2, paragraph 2** of the manuscript and presented the added zero-shot evaluation from **Figure 5 on Section 3.3**.

---

> > ### Author Response · Authors · 2026-01-25
> > **Response to Reviewer hqLM**
> >
> > **W1.2 (meta-learning):**
> >
> > With respect to the suggestion of comparing against meta-learned initializations, such as MAML, we agree that these are highly relevant initialization techniques and should be discussed in the manuscript. The reason for not including meta-learning for baseline comparison is two-fold.
> >
> > First, we want to note that we do not view our method as an alternative to meta-learning, or claim our prior is superior to priors obtained with meta-learning. In fact, we view meta-learning and population-based priors, such as the one proposed here, as complementary. A learned population prior could provide a strong starting point for meta-learning, potentially reducing the number of inner-loop updates, lowering data and memory requirements, and helping guide optimization toward more meaningful regions of the parameter space. We have therefore explicitly added this in the Discussion **(Section 4, last paragraph)**, as we hypothesize that combining meta-learning with our proposed biomechanically informed population prior could be an interesting future work.
> >
> > Second, in our setting, applying meta-learning directly presents with practical challenges. The INR represents a high-dimensional spatiotemporal displacement field defined over full 3D+time volumes, resulting in a large parameter space and significant memory requirements. Meta-learning methods typically require unrolling inner-loop optimization to (near) convergence and computing higher-order gradients, which in our case would require holding multiple large volumetric batches in memory simultaneously. This becomes particularly limiting when scaling the population sizes. A fair comparison to meta-learning would require a carefully designed training protocol and significantly higher computational resources, which we believe is beyond the scope of this revision.
> >
> > **W2 (comparison to VoxelMorph):**
> >
> > We agree that comparisons to learning-based registration methods such as VoxelMorph are valuable for benchmarking overall registration performance and for positioning the method within the broader landscape of image registration. Prior work has shown that INR-based, per-instance optimization approaches can offer advantages over discrete, grid-based CNN methods when modelling continuous motion (https://proceedings.mlr.press/v172/wolterink22a/wolterink22a.pdf). Our intention in this work, however, is to study the effect of a population-based initialization strategy on INR-based registration.
> >
> > In addition, a direct comparison in our setting is non-trivial. The proposed prior is learned from tagged–cine volumes and applied to cine-only registration using a continuous 4D representation, whereas methods such as VoxelMorph are trained end-to-end on discretized 3D volumes drawn from a fixed data distribution. Designing an equivalent training and evaluation setup would therefore require substantial adaptations beyond the scope of the revision.
> >
> > For these reasons, we focus our evaluation on comparing identical INR formulations with and without the proposed population prior, thereby isolating the contribution of the initialization strategy itself. We believe this design best supports the central aim of the paper, which is to demonstrate that meaningful population priors can be constructed for INRs and that such priors significantly improve convergence and performance in cardiac motion registration.
> >
> > **W3 (Jacobian determinant):**
> >
> > To interpret the increase in Jacobian determinant variability, we evaluated it separately within the left ventricle (LV) and myocardium, where the registration loss and regularization are applied, and in the remaining image regions, including the right ventricle (RV) and background **(Figure 3)**. This analysis shows that the Jacobian determinant within the LV and myocardium remains comparable between the prior-initialized and baseline models, whereas the observed degradation predominantly arises outside the regularized regions.
> >
> > The intuition behind this effect is that initializing the INR from a structured weight configuration reduces the network’s flexibility in early training, which can lead to variable deformations in regions that are less explicitly supervised or regularized. The LV myocardium, which is constrained during prior construction through a Jacobian regularization term, maintains comparable behaviour, whereas less constraint regions such as the RV and non-cardiac background exhibit increased variability (importantly, the folding ratio of the registration remains 0 for both methods across the image). We have added this analysis to the results in **Section 3.1, paragraph 2**.
> >
> > **W4 & W5 (missing values for the loss function and SIREN and mathematical notation):**
> >
> > Thank you for the feedback, we re-wrote the mathematical notation as suggested. We added the missing values for the regularization weight alpha and SIREN frequencies to the manuscript, as well as optimization details **(see last paragraph Section 2.1)**

---

> > > ### Comment · Reviewer_hqLM · 2026-01-26
> > >
> > > I'd first like to thank the authors for their response and for taking my considerations into account. I'd quickly like to comment on the response:
> > > - **W1**:
> > >   - I totally understand that applying *meta-learning* strategies - typically involving second-order optimization - to such high-dimensional data is challenging, and resource heavy. I appreciate that the authors include this into their discussion.
> > >   - *Regarding the prior*: I do understand some of your arguments. Starting with the same initialization and using the same architecture, all models start in the same "basin" of the loss landscape. With similar learning dynamics (optimizer, learning rate, etc. - that are now reported in the paper) the networks tend to stay in aligned regions within the weight space. However, this is not guaranteed! I checked some literature again and think there is papers like "Model soups" (https://arxiv.org/abs/2203.05482) that would support your claim. They fine-tune from a specific model and then average, which is somewhat similar to what you do. This, together with the added experiments (Figure 5) is much more convincing. If this is still feasible (I know the time is very short and I don't expect you to do this), a comparison to a prior constructed from INRs that all start with a different random initialization would be interesting. This would probably result in a large performance drop and you could nicely demonstrate why your approach is appropriate for solving the task. It is still interesting to see that such simple methods seem to work well - as demonstrated in your experiments.
> > >
> > > - **W2-W5**: Thank you for clarifying these points and for adding the additional experiments, rewriting the mathematical formulations and reporting relevant hyperparameters. I have no further comments or questions on these points.

---

> > > > ### Author Response · Authors · 2026-01-29
> > > >
> > > > Thank you for your response and interesting suggestion.
> > > >
> > > > As suggested, we additionally constructed a prior using an identical training setting but different random seeds for the  initialization for each patient.
> > > >
> > > > As you observed, a performance drop is expected in this setting, since parameters obtained from different random initializations are not aligned in weight space. Our results support this claim: the resulting prior network, when evaluated at zero-shot, produces a displacement field with near-zero magnitude.
> > > >
> > > > This behavior is consistent with the fact that Xavier initialization yields zero-mean, symmetrically distributed weights, such that independently trained INRs converge to solutions occupying different symmetric modes of the parameter space. As a result, direct parameter averaging could cause these independently trained weights to cancel out, yielding a low-magnitude displacement field as a starting point.
> > > >
> > > > We added this result in Appendix A, Figure 6 and discussed briefly on last paragraph of Section 3.3, and last paragraph of discussion Section 4.

---

### Official Review · Reviewer_q7oT · 2026-01-08

**Confidence:** 4
**Preliminary Rating:** 4
**Final Rating:** 4

**Summary:**

This paper proposes a tagged-informed population prior to accelerate and stabilize implicit neural representation (INR)–based cardiac motion estimation from cine CMR by leveraging high-fidelity deformation information from tagged CMR. Subject-specific INRs trained on tagged-cine data are aggregated into a population-level parameter prior, which is then used to initialize cine-only INRs, yielding faster convergence and more physiologically plausible motion. Experiments on UK Biobank and an independent multi-pathology cohort show that the prior halves optimization time while improving registration accuracy and producing smoother, more realistic myocardial strain curves

**Strengths:**

1 The manuscript is well written
2 The method is sound: Introduces a population-level initialization for INR-based cardiac motion estimation using tagged CMR
3 Demonstrates consistent reductions in optimization time with measurable gains in evaluation metrics

**Weaknesses:**

1 Unclear clinical value: The method is slow, and not trivial to deploy.
2 The paper is solid and technically sound, but it lacks elements that are particularly exciting to me.
3 Building a prior for registration is, "dangerous".

**Detailed Comments:**

Refer to the Pros and Cons.

**Justification Of Final Rating:**

This paper proposes a tagged-informed population prior to accelerate and stabilize implicit neural representation (INR)–based cardiac motion estimation from cine CMR by leveraging high-fidelity deformation information from tagged CMR. Subject-specific INRs trained on tagged-cine data are aggregated into a population-level parameter prior, which is then used to initialize cine-only INRs, yielding faster convergence and more physiologically plausible motion. Experiments on UK Biobank and an independent multi-pathology cohort show that the prior halves optimization time while improving registration accuracy and producing smoother, more realistic myocardial strain curves

Solid work
keep the original score

**Justification Of The Preliminary Rating:**

This paper proposes a tagged-informed population prior to accelerate and stabilize implicit neural representation (INR)–based cardiac motion estimation from cine CMR by leveraging high-fidelity deformation information from tagged CMR. Subject-specific INRs trained on tagged-cine data are aggregated into a population-level parameter prior, which is then used to initialize cine-only INRs, yielding faster convergence and more physiologically plausible motion. Experiments on UK Biobank and an independent multi-pathology cohort show that the prior halves optimization time while improving registration accuracy and producing smoother, more realistic myocardial strain curves

It's a good work, but not spectacular.

**Questions To Address In The Rebuttal:**

Not much.

---

> ### Author Response · Authors · 2026-01-25
> **Response to Reviewer q7oT**
>
> We thank the reviewer for their assessment and for recognizing the soundness of the proposed method and its consistent gains.
>
> (W1) Regarding clinical value and deployment, we agree that INR-based approaches remain computationally demanding compared to data-driven methods. Our primary goal in this work was to address a key bottleneck of INR-based cardiac motion estimation: slow per-subject optimization. By introducing the proposed prior, we show that needed optimization iterations are reduced by half while improving motion quality, as demonstrated by the qualitative strain study. We argue these advancements present a step toward enabling their deployment.
>
> (W2) We view this work as a promising building block towards more efficient and robust cardiac motion modelling. The proposed INR formulation makes it possible to aggregate subject-specific networks into a population-level initialization, without introducing additional model complexity. In the revised manuscript, we included a the zero-shot analysis **(Figure 5, and Section 3.3 last paragraph)**, where we show that the aggregated prior already encodes shared task-related structure at initialization, producing spatially aware displacement patterns before any optimization, while still allowing full flexibility during subject-specific training.
>
> (W3) We acknowledge the reviewer’s concern that building priors for registration can be dangerous. For this reason, we deliberately restrict the role of the prior to initialization only. All model parameters remain fully trainable, and the optimization is still driven by subject-specific data. Our results indicate that this strategy does not constrain the solution to a fixed motion pattern, as evidenced by generalization to unseen data from different institutions and pathologies. We have included in the revised version of the paper a clearer discussion of the sensitivity of the prior to cohort composition in Discussion **(Section 4, paragraph 3)**:
>
> *“[...] These findings indicate that the behavior of the prior is sensitive to the composition of the tagged cohort used during its construction. While a general population prior offers robustness and broad applicability, alternative strategies such as pathology-stratified priors or adaptive weighting schemes that emphasize specific motion patterns may better capture disease-specific mechanics without sacrificing generalization.”*

---

### Author Rebuttal · Authors · 2026-01-25

**Rebuttal:**

We have uploaded a (updated) revised version of the manuscript as Supporting Material. All changes and additions are highlighted in the document.

**Supporting Material:**

/attachment/0a1d697b982bfbaae9155c2d0d7d8ed630f72152.pdf

---

### Author Response · Authors · 2026-01-25
**Summary for area chair**

We thank the reviewers for their careful evaluation and constructive feedback. In response, we clarified the scope and assumptions of our method and strengthened the manuscript with additional analyses of the results and extended discussion.

This work introduces a tagged-informed population prior to accelerate and improve INR-based cine CMR registration. Across datasets and pathologies, the proposed prior consistently reduces convergence time while improving registration accuracy and motion quality.

Key reviewers’ concerns focused on the construction and interpretation of the prior. We expanded the Methods section to clarify why parameter aggregation is meaningful in this setting and added a zero-shot analysis showing that the aggregated prior encodes shared task-related structure at initialization.

Questions regarding deformation regularity were addressed through additional region-specific analyses of the Jacobian determinant. We show that increased Jacobian variability associated with the prior arises primarily outside the primary regularized myocardium area, while remaining stable in the left ventricle.

Finally, we expanded the Discussion section to explicitly address the sensitivity of the population prior to cohort composition, and to contextualize the qualitative strain analysis in the absence of tagged-derived strain ground truth.

---

### Comment · Area_Chair_uvcw · 2026-01-29
**Official Comment by Area Chair**

Dear Reviewers:

We kindly encourage you to take a moment to review the authors’ rebuttals and submit your feedback. Your prompt feedback is important for ensuring a thorough review. Thank you for your contributions to MIDL 2026. If you have responded to the authors' rebuttal, please feel free to ignore this message.

Thanks, AC

---

### Comment · Area_Chair_uvcw · 2026-02-01
**Official Comment by Area Chair**

Dear Reviewers:

Please take a moment to update their final rating by clicking “Edit” → “Official Review” and providing the Final Rating by February 1st 2026 (23:59 AoE).

Thanks, AC

---

### Meta-Review · Area_Chair_uvcw · 2026-02-07

**Recommendation:** Accept (Poster)
**Confidence:** 5

**Metareview:**

All reviewers found the proposed method to be novel and the results promising. The AC thus recommended acceptance of the paper.

---

### Decision · Program_Chairs · 2026-02-13

Accept (Poster)